# Risk Assessment Model for Offshore Wind Farm Decommissioning: Analysis of System Uncertainty, Risk Events, and Weather Delays

Emilio Macías-Amador<sup>1</sup>, Lissie de la Torre-Castro<sup>1</sup>, and Jonas Kaczenski<sup>1</sup>

<sup>1</sup>Fraunhofer Institute for Wind Energy Systems IWES, Am Seedeich 45, 27572 Bremerhaven, Germany

Correspondence: Emilio Macías-Amador (emilio.macias-amador@iwes.fraunhofer.de)

Abstract. With the ambitious goal of becoming climate neutral by 2050, the EU anticipates a significant expansion of offshore wind energy capacity. An offshore wind farm has an average lifespan of 20-25 years; therefore, the yearly number of turbines in need of decommissioning in the North and Baltic Sea will increase significantly starting in 2030. However, research on this phase is still being carried out and experience is limited. For this reason, this research work enables the development of a comprehensive risk analysis tool that simulates different OWF decommissioning strategies, focusing on critical cost and risk drivers. The methodology is focused on applying quantitative risk analysis to decommissioning project plans. Paired with Monte-Carlo simulations, this method can accurately assess the potential impact of various uncertainties, including systematic uncertainty and discrete challenges. By enhancing the understanding of decommissioning risks, the tool seeks to support smart planning and execution, ultimately contributing to cost reduction and improved viability of offshore wind energy. Findings show that delays derived from process risks can represent over 20% of project duration. Regarding discrete risk events, availability-related events are among the most significant, stressing the need for early scheduling of ports, vessels, and equipment.

### 1 Introduction

In the near future, the first commercial offshore wind farms (OWFs) installed in the North Sea will begin to reach their end of operation phase. For this reason, research on decommissioning is gaining momentum. The recent increase in installed capacity means that 1800 offshore wind turbines (OWTs) will reach their end of life before 2030 (Topham and McMillan, 2017), while 20,000 OWTs will do so between 2030 and 2040 (Topham et al., 2019). This will raise the significant challenge of safely removing thousands of artificial structures that are currently in the marine environment, at a time when vessels are already on high demand to meet installation deadlines.

Recent advances in simulation-based planning tools for offshore wind projects have primarily focused on the transport, installation, and operational phases. Tools such as discrete-event simulations and weather-window analyses are increasingly used to assess schedule risks under uncertain conditions. However, these risk models typically focus on weather-related delays and lack comprehensive integration of task-specific process risks, particularly in the context of decommissioning (Gintautas et al., 2016). Within Fraunhofer's ongoing Decommissioning minimization of Risks (DeMiR) research project, work was conducted to adapt the newest simulation-based, quantitative risk analysis models, and integrate them into a comprehensive risk analysis software to assess OWF decommissioning projects. This presents an opportunity to develop or adapt modeling

approaches that can quantify the influence of diverse risks on project duration, ultimately supporting more reliable and cost-effective decommissioning strategies.

#### 2 Methodology

The methodology employed for this work is based on ISO 31000:2018, a standard that provides a structured framework to assess risks and uncertainties. International standards for offshore wind energy recommend using it as a risk management framework (ISO 29400, 2020). Its goal is to support decision making by establishing a common language and workflow for identifying, analyzing, evaluating, and treating risks. In the context of OWF logistics, where complex marine operations intersect with weather-exposed activities. The proposed adapted methodology, consisting of five key stages, is presented in Figure 1:

Figure 1. Risk assessment methodology, adapted from [50].

#### 35 2.1 Setting the context

For the first two steps in the risk assessment methodology, the general activities, vessels and resources must be defined. The former include mobilisation for each vessel, transit to site, preparatory work, and unloading of components. After this, the next step in the process is to build a list with the decommissioning tasks for each OWF component, in line with the desired strategy. For the dismantling of the OWT, the task list is shown Table 1, together with the target duration for each individual step or the function used to calculate it.

**Table 1.** Example of tasks related to OWT removal.

| Subcomponent | Task                       | Duration (hrs)           | reference                      | repetitions per turbine |
|--------------|----------------------------|--------------------------|--------------------------------|-------------------------|
| -            | transit to site            | distance / vessel speed  |                                | 1                       |
| -            | positioning and jacking up | 6.0                      | (Schira Offshore Energy, 2022) | 1                       |
| -            | attaching traverse for OWT | $1.0^{a}$                |                                | 1                       |
| blade        | hook travel                | hub height / crane speed |                                | 3                       |
| blade        | slinging blade             | $0.5^{b}$                | (Schira Offshore Energy, 2022) | 3                       |
| blade        | separating blade           | $1.0^{b}$                | (Schira Offshore Energy, 2022) | 3                       |
| blade        | loading blade on deck      | hub height / crane speed |                                | 3                       |
| blade        | securing blades            | $0.5^{b}$                | (Nunemaker et al., 2020)       | 3                       |
| nacelle      | hook travel                | hub height / crane speed |                                | 1                       |
| nacelle      | nacelle slinging           | $2.0^{\rm b}$            | (Schira Offshore Energy, 2022) | 1                       |
| nacelle      | detaching nacelle from OWT | $3.0^{\rm b}$            | (Schira Offshore Energy, 2022) | 1                       |
| nacelle      | loading nacelle on deck    | hub height / crane speed |                                | 1                       |
| nacelle      | securing nacelle           | 2                        | (Nunemaker et al., 2020)       | 1                       |
| tower        | hook travel                | hub height / crane speed |                                | 1-3°                    |
| tower        | slinging tower section     | $2.0^{b}$                | (Myhr et al., 2014)            | 1-3°                    |
| tower        | removing tower section     | $3.0^{b}$                | (Myhr et al., 2014)            | 1-3°                    |
| tower        | loading tower section      | hub height / crane speed |                                | 1-3°                    |
| tower        | securing tower section     | 2.0                      | (Nunemaker et al., 2020)       | 1-3°                    |
| -            | traverse cut-off           | $1.0^{a}$                |                                | 1                       |
| -            | jacking down               | 1                        | (Schira Offshore Energy, 2022) | 1                       |
| -            | transit to next turbine    | distance / vessel speed  |                                | 1                       |

<sup>&</sup>lt;sup>a</sup> Assumed in this study.

This process of adding tasks with their duration is repeated for every turbine, and a similar approach is followed to add the activities related to the decommissioning of the remaining components. This includes adding a predefined duration based on the literature, or calculating them on a case-by case basis. The repetition of these steps results in a comprehensive list of tasks to dismantle every element in the OWF.

With a detailed list of times and resources identified, the next step requires the identification of risks at every step of the process. For this study, risks are grouped into two main categories: process risks and weather risks. The nature of the risk determines the approach used to quantify them and assess the potential effect of each source of uncertainty.

<sup>&</sup>lt;sup>b</sup> Estimated from the duration of complete component removal.

<sup>&</sup>lt;sup>c</sup> Exact value depending on tower structure.

https://doi.org/10.5194/wes-2025-195

Preprint. Discussion started: 15 October 2025

© Author(s) 2025. CC BY 4.0 License.

# 2.2 Associated process risks

The first type of risk identified are process risks. This category encompasses the concepts of systematic uncertainty and discrete risk events. This includes risks that stem from the planning and execution of the decommissioning project, as opposed to environmental conditions. Within this group, systematic uncertainty is a concept that can be used to categorise project delays that originate in earlier phases of the project (e.g., planning, consenting, design) but affect the execution stage indirectly (Prater et al., 2017). These uncertainties arise from structural or procedural shortcomings in project preparation.

Since their impact is broad and hard to quantify precisely, they are modelled by applying a variation to the baseline duration of each activity following a PERT distribution that ranges from a 10 % time reduction (e.g., overestimation, learning curve), to an increase of 50 % (e.g., multiple issues compounding). This approach captures uncertainty without requiring event-specific modelling, acknowledging that many factors affecting execution time are ingrained into the foundational assumptions of the project plan. The specific values were chosen based on similar works modelling risks in offshore wind projects (Garcia Munoz et al.).

The second type of risks that fall under this category are risk events. They refer to concrete events that have the potential to delay the OWF decommissioning process, ranging from a failure to properly detach one of the blades, to complete vessel machinery breakdown. The research work resulted in 15 risk events that might occur during OWF decommissioning projects. Beyond the mere description of the event, the research includes valuable information such as the likelihood of occurrence, as well as the optimistic, realistic and pessimistic impact on project schedule. These four parameters help to characterise the risk distribution, which becomes crucial when conducting iterative simulations. The breakdown of the 15 risks identified, along with their likelihood and estimated delays, is presented in Table 2.

To convert minimum, most likely and maximum delay estimations into a continuous function that allows for proper risk evaluation, PERT distribution is assumed for all the risk events included in this work. This type of probability distribution is commonly employed for events related to project management under uncertain circumstances (Parsi,2025; Reshi 2023). This process allows the model to take random samples of delays caused by different combinations of risk events, and create a distribution based on the updated duration of each activity.

# 2.3 Weather risk assessment

Once process risks are identified and analysed, the next step is to assess them together with weather risks. Offshore decommissioning activities are subject to harsh weather conditions which cause further project delays and uncertainties, leading to overrun costs. These costs pose in themselves the financial risk of having been unconsidered in the original projects provisions.

In order to estimate the consequences that the weather has on the project schedule duration, the modelling of the weather risk is based on the AdWaTSS method presented in (Lübsen and Wolken-Möhlmann, 2020) and the logistical concept is then simulated in the tool COAST, building on previous work for weather risk assessment (de la Torre-Castro et al., 2024; de la Torre-Castro and Kaczenski, 2025). Designed as a discrete-event simulator for offshore logistic processes, COAST takes into

**Table 2.** Risk event profiles for Monte Carlo simulation.

| Risk name                       | Event likelihood | Delay (hours) |             |         |
|---------------------------------|------------------|---------------|-------------|---------|
|                                 |                  | Minimum       | Most likely | Maximum |
| 1. Vessel availability          | 0.03             | 168           | 720         | 8760    |
| 2. Equipment availability       | 0.03             | 168           | 336         | 2160    |
| 3. Personnel availability       | 0.01             | 24            | 72          | 720     |
| 4. Port availability            | 0.05             | 12            | 24          | 168     |
| 5. Handling failure             | 0.001            | 3             | 12          | 336     |
| 6. Dropped object               | 3e-5             | 168           | 336         | 8760    |
| 7. Vessel machinery breakdown   | 8.5e-4           | 48            | 336         | 8760    |
| 8. Vessel collision             | 5.9e-5           | 48            | 336         | 8760    |
| 9. Contact with foundation      | 6.7e-5           | 48            | 336         | 8760    |
| 10. Jacking failure             | 0.01             | 2             | 6           | 8760    |
| 11. Tower detachment failure    | 0.01             | 3             | 5           | 4320    |
| 12. Nacelle detachment failure  | 0.02             | 3             | 10          | 4320    |
| 13. Blade detachment failure    | 0.04             | 3             | 19          | 4320    |
| 14. Foundation clearing failure | 0.04             | 4             | 16          | 4320    |
| 15. Foundation cutting failure  | 0.01             | 8             | 12          | 4320    |

consideration dependencies between activities, resources and the associated weather restrictions. The tool allows for detailed analysis of offshore operations, taking into account the vessels dependencies on weather and their implications at each step.

The mean results from the process risk modelling are utilised to update the original project schedule and obtain a distribution of delays related to weather. These delays are measured relatively by comparing the target end date of an activity and their new simulated end date, after consideration of risks. For statistical significance, the weather constraints of the required vessels are mapped into 43 years of time series of weather data. For this purpose, the ERA5 reanalysis data set (C3S, 2017) was selected. ERA5 is developed by the European Centre for Medium-Range Weather Forecasts and comprises hourly updates on climate variables including wind speed, significant wave height, and air temperature two metres above sea level. These sequence-based simulations capture the long-term variability and extreme events in weather conditions.

# 2.4 Reference scenario

To demonstrate the potential of the developed methodology, an exemplary analysis of a reference wind farm is thus presented. The reference offshore wind farm selected for this study is an average representation of the German OWFs currently installed in the North Sea. The wind farm comprises 80 turbines, each with a rated power of 5 MW, reflecting the typical scale of wind farms commissioned in the early-to-mid 2010s. Hub heights of 106 m and rotor diameters consistent with 5 MW-class turbines are considered. The turbines are mounted on monopile foundations, as the selected site is situated at an average water depth

of 30 m. The inter-array cable grid covers a length of 116 km. Eemshaven was selected as the base port for decommissioning operations, resulting in a travelling distance of 54 km. The wind farm is assumed to have been fully commissioned in 2015, with a planned decommissioning start date in March 1<sup>st</sup>, 2040, to be able to carry out the majority of the foundation cutting activities during the summer.

Five different vessels are considered for the decommissioning of the components: a jack-up vessel (JUV) will start by decommissioning the wind turbines, carrying back five turbines per trip. Once enough turbines are de-installed so that the vessel removing the inter-array cables can carry out its offshore works without waiting for more turbines to be dismantled, the decommissioning of the inter-array cables can start. The cables are exposed and re-winded into spools by a cable laying vessel (CLV) with a carousel weight of 9000 t. Afterwards, the offshore substation is dismantled in one trip by a heavy lift vessel (HLV) supported by a barge (B). The decommissioning of the foundations starts once all cables have been disconnected from the transition piece, a JUV equipped with an abrasive water jet cutting tool performs the process for five foundations per trip. Finally and once the foundations are dismantled, the scour protection layer is removed from the seabed using multipurpose vessel (MPV), with a dredger, and a barge to transport the stones back to shore. The operational restrictions of the vessels are presented in Table 3.

**Table 3.** Vessel operational weather restrictions used in the reference case.

| Vessel type | $\operatorname{Max} H_s [m]$ |           | Max wind speed $[m/s]$ |                   |  |
|-------------|------------------------------|-----------|------------------------|-------------------|--|
|             | Transit                      | Operation | Transit                | Operation         |  |
| JUV         | 3                            | 3         | 25                     | 15 (crane @100 m) |  |
| CLV         | 2                            | 2         | 20                     | 15 (@10 m)        |  |
| HLV + B     | 3                            | 2         | 25                     | 15 (@10 m)        |  |
| MPV + B     | 3                            | 2         | 25                     | 15 (@10 m)        |  |

# 2.5 Verification of the methodology

The risk modelling approach developed was internally tested to verify that the methodology described works as intended. To enable cross-validation of the scheduling concept, the decommissioning activities and their duration are included in this work. The COAST tool was validated in a bench-marking project organised by the National Renewable Energy Laboratory (NREL), based on the installation of an offshore wind farm (Shields, M. et al., 2021).

#### 3 Analysis of results

Without accounting for any delays, the expected completion of the decommissioning project is December 9<sup>th</sup>, 2040. While this corresponds to a project plan that spans over 283 calendar days, the days of effort (DoE) required are 443. This second metric considers the total hours spent among the different vessels, which are often working simultaneously at different locations. The comparison between these two metrics is useful to reflect the positive effect of conducting tasks in parallel whenever possible.

Figure 2. Target decommissioning project timeline.

Figure 2 illustrates the target project flow for the decommissioning of the reference wind farm, and the different time spans required to decommission each component. At this stage, the fastest element to decommission was found to be the offshore substation (OSS) with a calendar duration of 12 days. Next, the scour protection layer (SPL) and inter array cables (IAC) in the middle range, requiring 57 and 62 days respectively. The remaining components take over four months to decommission, with the OWT at 126 days and the foundation (FOU) at 183 days, making them the most time-consuming phases of the OWF decommissioning project.

# 3.1 Process risks

125

135

Following the established methodology, an uncertainty factor was included for every task, ranging from -10% to +50% to account for systematic errors. By repeating the same process for over 1000 iterations, the result is no longer a single DoE estimate, but a spread of possible values. On top of that process, the delay caused by discrete risk events is quantified, measuring the impact of 15 issues that can arise during the decommissioning (see Table 2). The emerging impact from these delays results in a broader distribution, given that the risks in this category often add several months to the project's schedule. The shape of the distribution is skewed to the left given the low probability of occurrence of most events, in most cases lower than 1%.

By bringing together both types of process risks, the simulation is able to provide a more robust percentile distribution of the total estimated project duration. This distribution accounts for the stochastic nature of both systematic uncertainty and risk event delays, varying based on their respective probability and impact. Figure 3 displays the resulting histogram of total DoE across iterations.

**Figure 3.** Histogram of total DoE considering process risks.

The median (P50) considering all process risks shows a DoE of 541 days, and an expected calendar duration of 372 days. This is equivalent to 0.93 days / MW. The optimistic estimate (P10) lays slightly below, at 347 calendar days, equivalent to 0.86 days / MW. On the other side of the spectrum, the conservative estimate (P90) is considerably higher. At 427 calendar days, it converts to a specific duration of 1.1 days / MW. Although these results does not include weather delays, the results are already comparable to literature from similar decommissioning projects as researched by (Topham and McMillan) (2017).

# 3.1.1 Risk Assessment Matrix

140

The fifteen discrete events defined in the model are evaluated through a Risk Assessment Matrix (RAM), depicted in Figure 4. The RAM enables the visualization of each risk based on its potential impact and likelihood of occurrence. Risks are plotted across two axes: the horizontal axis represents the estimated impact, measured in terms of potential delay in the project schedule, while the vertical axis captures the likelihood of the risk based on predefined probability distributions. Additionally, the size of each circle in the chart represents the uncertainty associated with the risk impact, allowing for a more intuitive interpretation of risk criticality.

The RAM identifies vessel availability as the most critical risk. Despite a relatively low likelihood of occurrence, its maximum potential delay (up to one year) and high uncertainty make it a priority concern. This result aligns with broader industry

155

Figure 4. Risk assessment matrix for risk events, numbered according to 2.

Table 4. Risk parameters for Monte Carlo simulation input.

| 1. Vessel availability        | 9. Contact with foundation      |
|-------------------------------|---------------------------------|
| 2. Equipment availability     | 10. Jacking failure             |
| 3. Personnel availability     | 11. Tower detachment failure    |
| 4. Port availability          | 12. Nacelle detachment failure  |
| 5. Handling failure           | 13. Blade detachment failure    |
| 6. Dropped object             | 14. Foundation clearing failure |
| 7. Vessel machinery breakdown | 15. Foundation cutting failure  |
| 8. Vessel collision           |                                 |

literature, which has identified vessel availability as a bottleneck in both installation and decommissioning phases due to limited supply and long lead times for replacement or chartering (Garcia Munoz et al.; Gatzert and Kosub, 2016).

Equipment availability follows in the high-priority zone, driven by a higher likelihood and considerable potential impact. Other moderate risks include port availability, foundation cutting failure, and blade detachment failure, all of which have meaningful consequences in both frequency and severity. Lower-ranking risks are generally characterized by extremely low likelihood of occurrence (e.g., dropped object, vessel collision) or more manageable impacts. However, their placement low on the matrix should not lead to complete dismissal, particularly for long or complex projects where compound effects may emerge.

160

# 3.2 Combined analysis: process and weather risks

The resulting distribution from total days of effort considering the mean results of the systematic uncertainties and risk events, together with the weather risk modelling is presented in Fig. 5. Given that weather data is limited to 43 years, less data points are available for this distribution, leading to a less continuous outline compared to Fig. 3. The DoE for the whole project considering both sources of delay is estimated at 760 days (1.9 DoE / MW) for the median scenario (P50), which corresponds to a 71 % increase from the target value.

Figure 5. Distribution of the days of effort for the project schedule, including process and weather risks.

Given the high-granularity of the model, the contribution of each type of delay can be further analyzed for different cases.

First, the P50 duration is dissected, which represents a value in the middle of all possible simulation outcomes.

Figure 6 details the contribution of type of delay in the decommissioning duration for each component, together with its target duration. At this percentile level, no major risk event delay is expected. Weather is the major source of additional delay, followed by system uncertainty.

For contrast, figure 7 show the average durations for each component, taking into account values that represent both the best and worst case scenarios. In this case, there are considerable risk event delays present, impacting mainly foundation and wind turbine decommissioning. The prominence of risk event delays stem from the fact that these events are unlikely, but have the potential to severely impact project schedule. The breakdown also makes it clear that components with longer

Figure 6. Breakdown of P50 project duration including systematic uncertainty (SU), risk events (RE) and weather delays.

**Table 5.** P50 delay comparison with process and weather delays.

|                                 | DoE [days] | $\mathrm{DoE}/\mathrm{MW}$ | Percentage of total |
|---------------------------------|------------|----------------------------|---------------------|
| Target project plan             | 443        | 1.1                        | 58%                 |
| Process risk delay <sup>a</sup> | 98         | $0.24^{b}$                 | 13%                 |
| Weather risk delay              | 219        | 0.54 <sup>b</sup>          | 29%                 |
| Total duration                  | 760        | 1.9                        | 100%                |

<sup>&</sup>lt;sup>a</sup> Process risks include both systematic uncertainty and risk events.

baseline durations accumulate more systematic uncertainty and weather delays, which is expected given that they are exposed to incidents for longer periods.

# 175 3.3 Sensitivity analysis

To expand on the insights extracted from the model, a sensitivity analysis was conducted varying only one of the risk events, recording its impact in total DoE. The selected variable was "foundation cutting failure", since it is related to a process with very limited experience; therefore, accurate likelihood of occurrence and impact of this event remains mostly uncertain (SeeOff Project Consortium, 2022).

In this exercise, both the probability of occurrence and the delay distribution parameters for the foundation cutting risk were systematically adjusted. To achieve this, the assumptions for minimum, most likely, and maximum delay were increased gradually, leading to a corresponding rise in the expected mean delay. This approach allowed for an exploration of how uncertainty

<sup>&</sup>lt;sup>b</sup> Can be interpreted as additional days of effort / MW.

Figure 7. Breakdown of average project duration including systematic uncertainty (SU), risk events (RE) and weather delays.

in both the frequency and severity of this risk event propagates through the model. The impact in project's DoE are summarized in Table 6.

**Table 6.** Sensitivity analysis of foundation cutting assumptions on project delay.

| Risk Level          | Likelihood | Mean Impact [days] | DoE [days] |
|---------------------|------------|--------------------|------------|
| Moderate (Baseline) | 0.01       | 30                 | 595        |
| High                | 0.02       | 45                 | 835        |
| Very High           | 0.03       | 60                 | 929        |
| Critical            | 0.04       | 75                 | 1047       |

The results reveal a strong sensitivity of the overall project duration to changes in probability and delay severity assumptions for foundation cutting failure risk. As the studied risk moves towards the yellow band of the RAM, the DoE rises by 8%. When tuning the parameters to position the risk event within the orange band, the DoE increases by 23%, and if the risk reaches the highest criticality level, the DoE becomes 43% larger compared to the baseline. This highlights the dominant role of foundation removal in decommissioning projects, and underscore the importance of standardizing the tools and strategies used for foundation removal.

# 4 Conclusions and future work

The core methodology was implemented for translating risk insights into probabilistic quantitative outputs, following the application of Monte Carlo simulations to model project duration under uncertainty. This approach was validated through

https://doi.org/10.5194/wes-2025-195

Preprint. Discussion started: 15 October 2025

© Author(s) 2025. CC BY 4.0 License.

200

205

the use of a reference offshore wind farm in the German North Sea, representative in size, turbine configuration, and site characteristics. By incorporating realistic parameters and expert-informed probability distributions, the model was able to evaluate the impact of individual risks on the overall schedule.

The results indicate that after weather risks, systematic uncertainty is the biggest driver of delay in most cases, while risk events play a role only in more conservative scenarios. From these discrete events, availability-related risks are among the most impactful, especially vessel availability. These are followed by handling-related failures and detachment issues, which, while less frequent, can result in major interruptions.

The importance of considering process and weather risks during the planning of decommissioning activities for an OWF has been presented in this paper. This is specially relevant in the coming years for a smooth transition between installation and decommissioning activities, given a shortage of the availability of required vessels. An OWF to be decommissioned located in the North Sea is used as reference case for the evaluation of uncertainties, risk events, and weather delays. The results show that considering these risks, the days of effort of all activities in the project may rise by 71 % compared to target.

Future work includes the extension of the risk catalogue in collaboration with industry experts, analysis of the uncertainty associated with vessel charter rates, and a validation of the simulation results with actual data from an OWF.

*Author contributions.* Macías-Amador and de la Torre-Castro developed the model code and performed the simulations. Macías-Amador prepared the manuscript with contributions from all co-authors. Kaczenski reviewed the manuscript.

210 Competing interests. The authors declare that they have no conflict of interest.

Acknowledgements. The work presented in this paper has been carried out within the research project DeMiR - Decommissioning Minimization of Risks [FkZ 03EE3096]. The project has received funding from the German Federal Ministry for Economic Affairs and Energy (BMWE) through the Projektträger Jülich (PtJ).

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
