# Peer review of "Risk Assessment Model for Offshore Wind Farm Decommissioning: Analysis of System Uncertainty, Risk Events, and Weather Delays"

_Wind Energy Science, 2025_

## Referee Comment (RC1)

**"Risk Assessment Model for Offshore Wind Farm Decommissioning: Analysis of System Uncertainty, Risk Events, and Weather Delays" (https://doi.org/10.5194/wes-2025-195)**

In this work, a probabilistic model to determine the duration of offshore wind farm (OWF) decommissioning is developed. The model considers general system uncertainty, rare risk events and delays due to weather conditions.

The idea of setting up a probabilistic model for OWF decommissioning is interesting and relevant, especially since many offshore wind turbines will reach the end of their service lifetime within the next decade. Considering uncertainties in this process is essential for meaningful results.

The paper is well written and nicely structured. However, it features several, three quite severe drawbacks. First, **the paper does not provide all data/information required to reproduce the results**. Second and even more important, **the entire uncertainty assessment seems to be based on very vague assumptions**. I hoped for some data provided by industry. However, data is either taken from literature (suitability is sometimes questionable (see comment 3) or parameter values are just assumed without sufficient justification. And third, the **overall innovation seems to be quite low**. The methodology is relatively similar to what has already been done for OWF installation. Surely, installation and decommissioning are two different things. However, if the underlying data is not based on real decommissioning data or at least industrial experience/expert knowledge but on the same assumptions made for installation, a case study will not yield very relevant findings. Hence, methodological innovation must be even higher, which is not the case here. Therefore, I cannot recommend the paper for publication without substantial improvements and extensions.

Further concrete comments are:

1) In line 21/22, you state that there is a lack of risk models that integrate task-specific process risks. This might be correct for the decommissioning. However, for the installation of OWF, there are such approaches and you use them. Hence, as installation and decommissioning involve quite similar tasks, the innovation of your approach is somehow limited. At least, you must discuss the available approaches for installation and highlight the differences.
2) Table 1: Tasks for the substation, cables, monopile etc. are missing. You should list all tasks. If there are too many tasks to be listed here, think about an appendix or some supplementary material. Furthermore, where did you get the data for the other tasks from?
3) L. 58: You state that the values and the distribution for the "systematic uncertainty" are taken from Garcia Munoz et al. (2023). However, it seems as if they have just chosen these values without any data foundation. Hence, is there any data or at least industry experience/expert knowledge? If not, the values are just educated guesses making your quantitative results less relevant. Furthermore, Garcia Munoz et al. (2023) investigate installations of OWF. Hence, the question is whether the same approach and the same values can be applied in case of decommissioning. If yes, the question is: how innovative is your work. If no, you should not use the values.
4) Table 2: Where does this data come from? And how reliable is it? Later, you conduct a parameter study for one of the values. However, as the uncertainty in these values is probably quite high, you should conduct a full sensitivity analysis for all parameters.
5) Table 2: Are the values the same for all vessel types? What about failures when decommissioning the substation or the cable?
6) Section 2.4: Stating the precise site of the reference wind farm would be nice to make your work more reproducible. I assume that you use the environmental conditions for exactly this site.
7) Section 2.4: Crane and vessel speeds are not mentioned, but needed for the simulation.

8) Section 2.5: It remains unclear, what you mean by "internally tested". Furthermore, you state that you want to enable cross-validation. However, if you want that other researchers can cross-validate your work, all data must be included (e.g., in the appendix or as supplementary material).

9) Section 3 and conclusions: How relevant is the entire case study if it is only done for one reference OWF and based on mainly "guessed" uncertainties? You should somehow show that the results are, at least to some extent, generally valid, i.e., for other sites, OWF and uncertainties.

10) L. 127: 1000 runs/samples (I would not call it iterations) does not seem to be a lot if your failure probabilities are sometimes $10^{-4}$.

11) L. 140/141: You state that the results are similar to what can be found in literature, although the weather delays are not yet included. My question is: why should they be similar? I assume that literature values do include weather delays. Does your model, after including weather delays, yield delays that are too high compared to literature values?

12) L. 159: Is it correct that you only use the mean results? If yes, why?

13) L. 160: How did you select the time period? Year 1-25, 2-25 etc. or randomly selected starting days? In the first case, I agree that you have only a few samples. In the latter case, you could generate more samples. On the downside, it would prevent you from always starting the decommissioning in March.

14) L. 169: With average, you mean the "arithmetic mean"? I would be very clear here, as you have already presented the P50 results.

15) Figure 6: Is it somehow included that systematic delays lead to further weather delays, e.g., an early systematic delay shifts the project to the winter which increases the weather delay, i.e., are interactions of the uncertainties considered? If yes, how are they counted in Figure 6? If no, why not?

16) Table 5: This table is neither discussed nor mentioned anywhere.

17) Section 3.3: Currently, you just conduct a parameter study for one parameter. I think that it would be interesting to see this for all parameters and perhaps even combinations of parameters in a real sensitivity analysis, e.g. based on elementary effects. This is especially important, as your uncertainty values are just educated guesses.

18) L. 193: You did not validate your approach, as it has not been tested for any real application. In line 207, you even state yourself that a validation is future work.

19) L. 195: How do we know that the parameters are realistic and probability distributions are expert-informed? In line 206/207, you write that future work should address topics like the incorporation of industry expert-knowledge and the validation using real data.

20) L. 206/207: In my opinion, these things should be already addressed in the current work. As the methodological innovation of this work is not that pronounced (see for example comment 1 or 3), at least the data source should be unique, i.e., include some real-world data. Currently, this is a major drawback of the work.

Typos etc.:

21) L. 5: Abbreviation "OWF" has not yet been explained.
22) L. 29 and 31: The cited standards are not on the list of references.
23) L. 32/33: "In the context…": The sentence structure is not correct.
24) Figure 1: "Adapted from [50]": What is "[50]"?
25) Table 1: All durations should be given with the same number of significant figures, e.g., 2.0, 0.50 etc.
26) L. 58 and 151: For "Garcia Munoz et al." the year is missing.
27) Table 4: If I am not mistaken, it is just a repetition of Table 2 and can be removed.
28) L. 169: "Fig. 7" not "figure 7".
29) References: Some references seem to be incomplete, e.g., Garcia Munoz et al. or Shields et al.